# The Complex Relationship between Diabetic Retinopathy and High-Mobility Group Box: A Review of Molecular Pathways and Therapeutic Strategies

**DOI:** 10.3390/antiox9080666

**Published:** 2020-07-26

**Authors:** Marcella Nebbioso, Alessandro Lambiase, Marta Armentano, Giosuè Tucciarone, Vincenza Bonfiglio, Rocco Plateroti, Ludovico Alisi

**Affiliations:** 1Department of Sense Organs, Faculty of Medicine and Odontology, Policlinico Umberto I, Sapienza University of Rome, p. le A. Moro 5, 00185 Rome, Italy; marcella.nebbioso@uniroma1.it (M.N.); marta.armentano@uniroma1.it (M.A.); giosue.tucciarone@uniroma1.it (G.T.); rocco.plateroti@uniroma1.it (R.P.); ludovico.alisi@uniroma1.it (L.A.); 2Department of Ophthalmology, University of Catania, Via S. Sofia 76, 95100 Catania, Italy; vincenzamariaelena.bonfiglio@unipa.it

**Keywords:** antioxidants, diabetes mellitus, diabetic retinopathy, free radicals, high-mobility group box 1 (HMGB1), inflammatory pathways, novel therapies, oxidative stress

## Abstract

High-mobility group box 1 (HMGB1) is a protein that is part of a larger family of non-histone nuclear proteins. HMGB1 is a ubiquitary protein with different isoforms, linked to numerous physiological and pathological pathways. HMGB1 is involved in cytokine and chemokine release, leukocyte activation and migration, tumorigenesis, neoangiogenesis, and the activation of several inflammatory pathways. HMGB1 is, in fact, responsible for the trigger, among others, of nuclear factor-κB (NF-κB), tumor necrosis factor-α (TNF-α), toll-like receptor-4 (TLR-4), and vascular endothelial growth factor (VEGF) pathways. Diabetic retinopathy (DR) is a common complication of diabetes mellitus (DM) that is rapidly growing in number. DR is an inflammatory disease caused by hyperglycemia, which determines the accumulation of oxidative stress and cell damage, which ultimately leads to hypoxia and neovascularization. Recent evidence has shown that hyperglycemia is responsible for the hyperexpression of HMGB1. This protein activates numerous pathways that cause the development of DR, and HMGB1 levels are constantly increased in diabetic retinas in both proliferative and non-proliferative stages of the disease. Several molecules, such as glycyrrhizin (GA), have proven effective in reducing diabetic damage to the retina through the inhibition of HMGB1. The main focus of this review is the growing amount of evidence linking HMGB1 and DR as well as the new therapeutic strategies involving this protein.

## 1. Introduction to Diabetic Retinopathy (DR)

Diabetes mellitus (DM) is a well-known metabolic disease that causes numerous chronic complications. To this day, the number of patients affected by type one and type two DM is estimated to be 463 million. This number will rise to 700 million by 2045 (around 10.9% of the global population) [1]. The most common complications, such as diabetic retinopathy (DR), are caused by microvascular damages. Nowadays, the number of patients affected by DR is calculated as around 93 million [2]. These numbers place DR as the fifth most common cause of severe visual impairment in the world [3]. From a clinical point of view, DR is characterized by typical vascular and macular abnormalities. In non-proliferative DR (N-PDR), the most common findings are microaneurysms, cotton wool spots, hemorrhages, hard exudates, and venous dilatation. The progression toward the stage of PDR is defined by the development of neovascularization that may lead to retinal and vitreous hemorrhages, fibrovascular proliferation, and tractive retinal detachment. Other complications of DR are neovascular glaucoma, steaming from iris neovascularization, and macular edema (Figure 1 and Figure 2) [4,5].

The pathogenesis of DR is an extremely complex mechanism that involves numerous biochemical and inflammatory pathways triggered by long exposition to hyperglycemia. The development of DR is characterized by the concomitant participation of vascular endothelial dysfunction, pericyte loss, and neurodegeneration, which ultimately leads to hypoxia and neovascularization [6]. Interestingly, neuronal degeneration appears to precede vascular disease and develop as an independent mechanism [7].

The persistence of high levels of blood glucose is determinant for the activation of inflammatory mechanisms, the enhancement of oxidative stress, and, consequently, the production of advanced glycation end-products (AGEs) [4]. The inflammation determines the local accumulation of cytokines, such as vascular endothelial growth factor (VEGF), tumor necrosis factor-alpha (TNF-⍺), and inducible nitric oxide synthase (iNOS), that favor the establishment of hypoxia in the diabetic retina [8]. Inflammation leads also to the accumulation of chemokines and adhesion molecules such as intercellular adhesion molecule-1 (ICAM-1). This mechanism causes the migration of leukocytes towards the retinal endothelium, increased vascular permeability, and the breakdown of the blood–retinal barrier (BRB) that ultimately leads to edema [9].

Hyperglycemia determines the production of AGEs. The binding of AGEs to their receptors (RAGE) allows for the activation of nuclear factor-κB (NF-*κ*B) pathways, which ultimately leads to the production of reactive oxygen species (ROS) and the reduction of antioxidant defense systems [10]. Oxidative stress conducts to the metabolic memory phenomenon in mitochondria. This phenomenon is deemed responsible for the persistence of vascular damage, even when glycemic control is perfectly achieved [11]. Moreover, hyperglycemia determines the production of sorbitol through the activation of the polyol pathway. High volumes of sorbitol lead to the depletion of reduced glutathione (GSH) and the accumulation of ROS [12].

Hyperglycemia is also able to activate different isoforms of protein kinase C (PKC). This enzyme participates in the retinal vascular damage through its efficiency in the induction of nicotinamide adenine dinucleotide phosphate oxidase (NOX). The resulting O_2_^−^ contributes to the worsening of endothelial dysfunction [13,14]. PKC is also involved in the increase of endothelial cell death and pericyte loss via the accumulation of oxidative and nitrosative products, contributing to the development of microaneurysms and the recruitment of leukocytes [15].

Another pathway heavily affected by chronic exposition to hyperglycemia is the hexosamine pathway. The alternative cycle to glycolysis is needed to convert the excess of fructose 6-phosphate that cannot be metabolized by classic glycolysis. This leads to the production of *N*-acetyl glucosamine and the overexpression of transforming growth factor-β1 (TGF-β1) and plasminogen activator inhibitor-1 (PAI-1), increasing the apoptotic rate of endothelial cells (EC) and pericytes [16,17]. Moreover, AGE accumulation activates the hexosamine pathway, determining the production of angiopoietin-2 and the development of neovascularization [18,19]. Lastly, it has been recently demonstrated that the activation of the innate immune response facilitates the development of inflammation and therefore DR. Specifically, concentrations of toll-like receptor (TLR)−4 and −2 as well as their downstream inflammatory cytokines TNF-α, interleukin (IL)-1β, and interferon (IFN)-β were found to be significantly increased in murine models of DM [20,21].

Current therapeutic approaches, such as anti-angiogenic agents or corticosteroids intravitreal injections and laser therapy, only target the manifestations of DR. The complexity of the metabolic pathways activated during DR shows how single-target therapies have limited success [22,23,24]. The lack of preventive treatments and the increasing number of patients show the need for the development of new specific agents targeting the metabolic pathways that lead to DR (Scheme 1).

## 2. Introduction to High-Mobility Group Box 1 (HMGB1)

High-mobility group (HMG) proteins are a group of non-histone nuclear proteins discovered in 1973 in the calf thymus, including three families, named HMGB, HMGN, and HMGA [25]. This group of proteins owes their name to their high electrophoretic mobility [26]. The high-mobility group box (HMGB) family contains several different proteins unified by the constant presence of at least one HMGB. The most studied and ubiquitary protein of this family is HMGB1 [27]. HMGB1 is an evolutionary conserved chromatin-binding protein composed of 215 amino acids and characterized by two DNA binding domains named Box-A and Box-B and a C-terminal acidic domain [28]. Initially considered a nuclear protein, HMGB1 has subsequently shown a cytosolic location inside several cell organelles and structures, such as mitochondria and the cellular membrane as well as extracellular space [29].

HMGB1 may be present in a reduced, oxidized, or disulfide form. Its actions appear to be largely dependent on the redox state [30]. The reduced form is characterized by the reduction of specific cysteine residues. In this configuration, HMGB1 can recruit leukocyte independently from the release of cytokines or chemokines [31]. The oxidized configuration determines the loss of the immunogenic properties of HMGB1 [32]. Lastly, the disulfide form activates the NF-κB inflammatory pathway, determining the production of IL- 6, -8, and TNF-⍺ [31].

HMGB1, due to its ubiquitous location in the cell, performs numerous activities (Figure 3).

Inside the nucleus, HMGB1 controls chromatin stability and replication, nucleosome release from damaged cells, gene recombination, and transcription, DNA repair, and replication [27,33].

Cytosolic HMGB1 is usually secondary to the shuttling of nuclear HMGB1 in response to hypoxia, chemokines, cytokines, and ROS. In the cytosol, HMGB1 acts as a positive regulator of autophagia [34]. HMGB1 expression on the surface of cellular membranes is responsible for the activation of innate immunity and mediates cellular adhesion [35,36].

Extracellular HMGB1 is involved in numerous activities such as the regulation of T-cells [37], stem cells [38], and neoplastic cell differentiation [39]. This protein is also involved in the management of the inflammatory response, through the activation of numerous different immune cells [40,41], and the promotion of cytokine release [42,43]. HMGB1’s extracellular functions consist of cellular proliferation [44] and migration [45], including vascular growth during inflammatory or neoplastic diseases and tissue repair [46,47]. During the inflammatory response, HMGB1 is secreted by macrophages, platelets, EC, and monocytes, as well as necrotic or damaged cells [48]. Disulfide HMGB1 binds together with myeloid differentiation factor-2 and TLR-4, determining the formation of a complex that triggers the inflammatory response [49,50]. In addition, HMGB1 deficient cellular lines show a reduced capacity to induce cytokines [51]. The binding of HMGB1 to RAGEs determines the formation of a complex responsible for the activation, among others, of NF-κB, phosphatidylinositol 3-kinase (PI3K)/PKB, mitogen-activated protein kinase (MAPK), and TNF-α pathways [52,53,54]. Thus, HMGB1 is involved in myriad diseases, such as hypoxia-induced injury [55], microglial damage and neuroinflammation [56], vascular barrier damage [57], and inflammatory heart diseases [58]. Moreover, ROS, through the activation of the NF-κB pathway, are responsible for the passive and active secretion of HMGB1 in monocytes and macrophages [59]. HMGB1 is recognized to be a direct angiogenic molecule as it induces a pro-angiogenic phenotype in EC [60,61]. It can, moreover, stimulate angiogenesis through the activation of the MAPK/extracellular signal-regulated kinase (ERK) 1/2 pathway. The bond between HMGB1 and RAGE results in the stimulation of NF-κB signaling in leukocytes, which leads to the production of proinflammatory and angiogenic molecules [62]. HMGB1 in conjunction with TLR-4 can influence the development of neovasis in proliferative and metabolic diseases [63,64]. Moreover, it has been demonstrated that HMGB1 can mediate angiogenesis through the activation of hypoxia-induced factor-1α (HIF-1α) [65].

In conclusion, HMGB1 shows a wide range of interactions in both physiological and pathological mechanisms. The next section of the review will focus on the growing amount of evidence linking HMGB1 expression and the development of DR. The main focus of this review is the growing amount of evidence linking HMGB1 and DR as well as the new therapeutic strategies involving this protein.

## 3. HMGB1 and DR

At the moment, information regarding the function of HMGB1 in DR is mostly limited to murine models and in vitro studies. DM upregulates the expression of HMGB1, leading to the activation of inflammatory signaling pathways such as the RAGE-mediated activation of ERK1/2-NF-κB. Intravitreal injection of HMGB1 mimics the effects of diabetes and increases RAGE, ERK1/2, NF-κB, and proinflammatory biomarkers such as ICAM-1 and soluble ICAM-1 (Scheme 2). These mechanisms decrease TLR-2 and occludin expression, increasing retinal vascular permeability and disrupting the stability of tight junction complex between adjacent retinal microvascular EC [66].

High glucose stimulates the translocation of HMGB1 into the cytoplasm of retinal pericytes. RAGEs act as receptors for HMGB1 and, in diabetes, their expression is enhanced. HMGB1 is involved in the induction of DR through the activation of this receptor. HMGB1, through the binding of RAGEs, enhances the transcriptional activity of NF-κB in retinal pericytes in in vitro and in vivo models. Hyperglycemia also increases the binding of NF-κB to the RAGE promoter, inducing the overexpression of RAGEs and therefore establishing a vicious cycle [67].

HMGB1 is strictly related to the signal transducer and activator of transcription-3 (STAT-3). Constant intake of HMGB1 inhibitor glycyrrhizin (GA) attenuates the upregulation of phosphorylated STAT-3 (pSTAT-3). The inhibition of STAT-3 blocks HMGB1-induced VEGF upregulation and human retinal microvascular endothelial cell (HRMECs) migration, suggesting the role of STAT-3 in mediating HMGB1-induced angiogenesis in DR [68].

HMGB1 induces the significant upregulation of IL-1β and ROS and the expression of NOX2, caspase-3, and poly ADP-ribose polymerase-1 (PARP-1) in HRMECs [69].

HMGB1 may have a role in the alteration of BRB HMGB1 expression, which is enhanced in the retinas of diabetic rats, and BRB permeability is significantly increased [70].

Sirtuin 1 (SIRT1) is a member of the SIRT family of proteins with deacetylase activity. Many studies report its role in DNA repair, oxidative stress, angiogenesis, inflammation, and senescence. There is a strong link between SIRT1 expression and the development of DR and PDR. In particular, hyperglycemia and diabetes cause the downregulation of SIRT1, thus resulting in inflammation, angiogenesis, an increase in oxidative stress, and vascular permeability, all of which are hallmarks of diabetic damage [71]. There is a functional link between HMGB1 and SIRT1 in the regulation of the diabetes-induced breakdown of the BRB. Intravitreal injection of HMGB1 in normal rats results in the downregulation of SIRT1. The HMGB1 inhibitor GA attenuates the downregulation of and normalizes retinal SIRT1 expression. Moreover, treatment with the SIRT1 activator resveratrol attenuates the diabetes-induced downregulation of SIRT1, accompanied by reduced expression of HMGB1 and RAGEs. Resveratrol may confer protection against the diabetes-induced breakdown of BRB through SIRT1 upregulation and HMGB1 downregulation [72]. HMGB1, insulin-like growth factor-binding protein 3 (IGFBP-3), SIRT1, and protein kinase A (PKA) are strictly related. IGFBP-3 increases SIRT1 and decreases HMGB1. PKA mediates the reduction in cytoplasmic HMGB1 by increasing IGFBP-3 and SIRT1 activities [73].

Chen et al. found increased expression of HMGB1 and its receptor RAGEs TLR-2 and TLR-4 in the retinas of type 2 diabetic rats and human retinal pigment epithelial cell line-19 (ARPE-19) exposed to high glucose. The NF-κB activity was found to be increased as well. The blockage of HMGB1 downregulated NF-κB hyperactivation and VEGF production in high glucose cultured ARPE-19 cells [74].

High levels of HMGB1 expression are due to both gene transcription and protein synthesis. The specific mechanism by which HMGB1 leads to DR is unclear. It may exert its function via the TLR-9 pathway. The expression of TLR-9 was increased and positively related to the expression of HMGB1 [75].

A high glucose environment could promote HMGB1 expression and activate TLR-4 and NF-κB overexpression in retinal ganglion cells (RGC), thus leading to the inhibition of cell survival and growth. TLR-4 is an important receptor for HGMB-1 that is largely expressed in the nervous system and can regulate neuron growth and proliferation. When HMGB1 binds to TLR-4, it activates several signaling pathways such as NF-κB with the release of inflammatory cytokines, chemokines, and colony-stimulating factors, leading to leukocyte adhesion and inflammation [76]. Yu et al. showed a higher expression of HMGB1 in diabetic rats associated with the upregulation of phospholipases A2 (PLA-2), TNF-⍺, VEGF, and ICAM-1. Regarding HMGB1 receptors, RAGEs protein was increased, whereas TLR-1 was reduced, suggesting that HMGB1 effects are RAGE-mediated [77].

Injury and death of the retinal pericytes and EC in DR might be due to the HMGB1/PLA2 induced cytotoxic activity of glial cells as well as the direct effect of HMGB1 on EC. HMGB1 could mediate EC death directly and pericyte death indirectly through the HMGB1-induced cytotoxic activity of glial cells. Regarding PLA2 it seems to be a positive regulator of VEGF-induced angiogenesis [78].

HMGB1 has an important role in angiogenesis. It can act directly through RAGEs and TLR-4 with EC activation, proliferation, and migration. HMGB1 also promotes angiogenesis indirectly through the production of proangiogenic cytokines, such as VEGF, TNF-⍺, and IL-8 from EC and activated macrophages [79]. The same role of HMGB1 was also demonstrated by Santos et al. The authors suggest that HMGB1 is not able to mediate angiogenesis in the retina by itself [80].

According to Lee et al., AGEs cause a rise in intracellular ROS, inducing the release of HMGB1 into extracellular space. HMGB1 augments the signal via RAGEs or TLR and mediates the secretion of VEGF-A through the c-Jun N-terminal kinases signaling pathway that was blocked by HMGB1 inhibitor GA. This could be a possible way through which HMGB1 upregulates VEGF [81].

HMGB1 and VEGF-A expression are upregulated in serum samples of DR patients and are positively associated. The in vitro up-regulation of HMGB1 inhibits the retinal pigmented epithelium (RPE) cell viability and induces apoptosis. HMGB1 administration to RPE cells in high glucose conditions up-regulates the expression of VEGF-A [82].

The silencing of HMGB1 inhibits the activation of MAPK and NF-κB signaling pathway; modulates the levels of VEGF, ICAM-1 and vascular cell adhesion molecule-1 (VCAM-1), therefore influencing endothelial permeability; attenuates cell apoptosis, BRB damage, and the inflammatory response induced by high concentration of glucose [83].

HMGB1 may inhibit the expression of NF-κB light polypeptide gene enhancer in B-cell inhibitor-α, a protein capable of inhibiting NF-κB by binding to its promoter region. This determines the activation of the NF-κB pathway, influencing inflammation and angiogenic processes, thus leading to DR. High levels of HMGB1 stimulate apoptosis and inhibit the proliferation of human retinal endothelial cells (HRECs). HMGB1 may determine apoptosis through the NF-κB pathway thanks to an alternate mechanism of non-perfusion and neovascularization [84].

There is a potential link among HMGB1, vascular adhesion protein-1 (VAP-1), oxidative stress, and heme oxygenase-1 (HO-1) in the pathogenesis of inflammation and angiogenesis associated with PDR. HMGB1 levels are consistently increased in the vitreous of patients with PDR, particularly higher in patients with active PDR. Exogenous HMGB1 activates HRMECs to upregulate the adhesion molecule ICAM-1.

Increased levels of the oxidative marker 8-hydroxydeoxyguanosine (8-OHdG) in the vitreous of PDR patients, particularly in active PDR, have been found. The positive correlation between vitreous levels of HMGB1 and 8-OHdG in HRMECs suggests that HMGB1 is associated with oxidative stress.

Regarding VAP-1, there was a significant correlation between the levels of sVAP-1, HMGB1 concentration, and 8-OHdG in vitreous. Expression of VAP-1 was higher in diabetic patients compared to controls in the RPE, whereas no significant difference was found in the neuroretina.

Stimulation with HMGB1 caused the upregulation of HO-1 in HRMECs. HO-1 levels were significantly higher in eyes with active neovascularization compared with eyes with involuted PDR. These findings suggest that HO-1 might contribute to PDR angiogenesis and progression. Moreover, VEGF can induce the expression of HO-1 that stimulates the synthesis of VEGF in a positive feedback loop [85].

Vascular EC and stromal cells in diabetic epiretinal membranes express HMGB1, RAGE, osteopontin (OPN), and early growth response protein-1 (Egr-1). In diabetic epiretinal membranes, these proteins and receptors are specifically localized in myofibroblasts. This suggests that HMGB1/RAGE/OPN/Egr-1 signaling pathway is involved in the inflammatory, angiogenic, and fibrotic responses in proliferative vitreoretinopathy (PVR) and may contribute to the instauration of PDR and its most dangerous complications [86].

OPN, HMGB1, and connective tissue growth factor (CTGF) were upregulated in the vitreous of patients with PVR, particularly in their active form, whereas increased levels of pigment epithelium-derived factor (PEDF) may be a response designed to counteract the activity of the angiogenic and fibrogenic factors during the progression of PDR and PVR [87].

There is a relationship between the activity of PDR, the presence of vitreous hemorrhages, and levels of HMGB1. In fact, HMGB1 is higher in patients with active PDR compared with inactive PDR and is higher in PDR patients with vitreous hemorrhages compared with patients without it [88].

Shen et al. found that HMGB1, VEGF, RAGE, and IL-1β levels were significantly elevated in the vitreous and serum of patients with PDR, suggesting that the upregulation of HMGB1 might contribute to the initiation and progression of angiogenesis in PDR and that the HMGB1/RAGE signaling axis has a role in the progression of PDR [89].

The upregulation of HMGB1 can induce the downregulation of brain-derived neurotrophic factor (BDNF), a neurotrophin with a neurogenetic function, and also of synaptophysin, an integral membrane protein of synaptic vesicles involved in neurotransmission. HMGB1 upregulates cleaved caspase-3 in vitreous fluid and serum from patients with PDR, as well as in the retinas of diabetic rats. HMGB1 inhibitor GA is able to revert the downregulation of BDNF.

RAGEs and ICAMs levels are upregulated in the serum of patients with PDR. RAGEs bind its ligands, preventing their link to RAGE, therefore blocking the inflammatory cascade. Elevated levels of RAGEs in the serum of patients with PDR could negatively regulate inflammation and limit diabetes-induced retinal vascular and neuronal dysfunction [90].

HMGB1 and VEGF levels were higher in vitreous from PDR patients. Moreover, there were increased levels of soluble vascular endothelial-cadherin that could be a marker of EC activation or injury associated with angiogenesis, inflammation, and the breakdown of the inner BRB. Finally, there was lower angiogenic activity in patients with higher levels of soluble endoglin, suggesting that it could be protective against pathological angiogenesis [91].

The intravitreal injection of HMGB1 in normal rats mimics the effect of DM, with increased expression of HMGB1 protein and mRNA, caspase 3, and levels of glutamate (responsible for excitotoxic neuronal death). HMGB1 inhibitor glycyrrhizic acid attenuates all of these effects. The early retinal neuropathy induced by diabetes is, at least in part, attributable to the diabetes-induced upregulation of HMGB1. Inhibiting the release of HMGB1 with a constant intake of GA results in the reduction of diabetes-induced retinal neuropathy. This could be a novel therapeutic approach to DR. [92].

The induction of DM and intravitreal injection of HMGB1 in normal rats resulted in the significant upregulation of HIF-1α, Egr-1, tyrosine kinase 2 (TYK2), and the CXCL12/CXCR4 chemokine axis. HIF-1α is associated with retinal inflammation induced by diabetes, Egr-1 may play a role in the development of vascular complications of DM, and the CXCL12/CXCR4 chemokine axis contributes to neovascularization. All these upregulations are mediated by the interaction of HMGB1 with RAGE. Inhibition of the release of HMGB1, for example with GA, attenuates the upregulation of all these molecules [93].

Exposure to hypoxia is able to release HMGB1 from RPE cells. HMGB1 may stimulate the overproduction of angiogenic and fibrogenic factors such as VEGF and CTGF in RPE cells. HMGB1 is involved in DR pathogenesis through binding to TLR-4, RAGE, and their signaling cascades such as PI3K, p38/MAPK, and NF-κB [94,95].

## 4. Future Therapeutic Approaches

Numerous molecules have been studied as inhibitors of HMGB1 in recent years for the treatment and prevention of DR and its complications (Table 1).

### 4.1. Glycyrrhizin (GA)

GA is a triterpene glycoconjugate naturally extracted from licorice root (*Glycyrrhiza glabra*). It is composed of two molecules of glucuronic acid and glycyrrhetinic acid aglycone [96]. This molecule inhibits the chemotactic and pathogenic functions of HMGB1 by binding directly the A and B boxes [97]. GA shows a wide range of effects such as antibacterial, hepatoprotective, antiproliferative, antiallergic, and antiviral [98].

As a result of recent studies on ARPE cells, it has been demonstrated that HMGB1 is connected to the increase in angiogenesis and fibrosis during the course of DR [94]. Oral administration of GA in diabetic mice strongly inhibited HMGB1 concentration in retinas. This result led to a reduction in vascular and neuronal damage related to DR. The anti-inflammatory effects of GA were mediated by the inhibition of TNF-⍺, IL-1β, and the cleavage of caspase-3 in retinal EC. GA, through the inhibition of HMGB1, reduces ROS concentrations and blood circulating glucose [99]. In another work, GA reduced TLR-4 concentrations and ischemia-reperfusion damage as well as increasing the expression of insulin receptors, partially preserving the anatomical integrity of the retina [100]. In a recent study, Liu et al. demonstrated that exchange protein for cAMP1, an inflammatory molecule involved in leukostasis, acts in synergy with GA. The combination of these two proteins strongly inhibits HMGB1 through the activation of SIRT1. SIRT1 deacetylates HMGB1, exerting a protective role in the diabetic retina [101]. GA also suppresses the proangiogenic effects of HMGB1 as it blocks AGE-induced upregulation of VEGF [81]. Abu El-Asrar and Mohammad’s workgroup demonstrated, in a diabetic murine model, that GA can inhibit HMGB1’s cytokine-like activities. Specifically, oral GA determines a reduction in HIF-1α, transcription factor Egr-1, TYK2, CXCL12, and CXCR4 [93]. Moreover, the same authors demonstrated that GA can inhibit the upregulation of STAT-3 induced by HMGB1 and its translocation in retinal Müller cells [68], upregulate BDNF expression in experimental mice [90], attenuate the expression of NOX2, caspase-3, and PARP-1 in the retinas as well as lowering the concentrations of ROS [69] and cleave caspase-3 glutamate and downregulating neurodegeneration mediators and markers in murine retinas [92] and attenuating the expression of retinal ICAM-1 [84]; lastly, it inhibits HMGB1 mediated activation of NF-κB. [66]. GA in association with resveratrol shows the ability to replenish retinal SIRT1 expression [72]. 

All this evidence points in the direction of the potential use of GA in the prevention of DR and its complications. It is worth mentioning that, in human studies on male patients affected by chronic hepatitis and type 2 diabetes, the administration of GA reduced serum testosterone aggravating insulin resistance, atherosclerosis, and sexual dysfunctions [102]. Further studies, especially on human subjects, are needed in order to confirm the pathways and molecules involved and their efficacy and safety.

### 4.2. Small Interfering RNAs/Short Hairpin RNA (siRNA/shRNA) 

Small interfering RNAs (siRNAs) are a class of double-strand RNA usually constituted by 21–25 nucleotides that are gaining importance as therapeutic tools in numerous diseases. SiRNAs are capable of selectively binding specific genomic sequences, silencing them and therefore inhibiting the protein expression [103,104].

SiRNA HMGB1 transfection can repress HMGB1 RNA overexpression, determining the suppression of TLR-4 and NF-κB mRNA in RGCs. The downregulation of these inflammatory pathways can promote the survival and growth rates of RGCs [76]. A similar study conducted by Jiang and Chen confirmed these results in both in vivo and in vitro models. HMGB1 suppression, mediated by intravitreal injections of siRNA, is capable, in diabetic rats, of reducing retinal apoptosis rates as well as improving retinal function. In HRECs exposed to high glucose concentrations, siRNA HMGB1 improved cell viability and reduced the oxidative damage lowering ROS production [105]. The same study group demonstrated the protective role of HMGB1 inhibition in murine DR models. Retinal cells isolated from 8-year-old rats were incubated with a recombinant lentivirus containing short hairpin RNA (shRNA) for HMGB1. Through this mechanism, the authors obtained the silencing of HMGB1 gene expression. The results showed the downregulation of both MAPK and NF-κB, contributing to the reduction of inflammation, cell death, and BRB breakdown [83].

### 4.3. Polygonum Cuspidatum (PCE)

*P. cuspidatum*, also known as “Hojang-geun” in Korea, is a commonly employed herbal medicine in East Asia. The plant shows anti-inflammatory and anti-diabetic effects [106]. Recent works have explored the potential role, as a preventive treatment, of *P. cuspidatum* extract in diabetic nephropathy [107]. PCE is rich in resveratrol, polidatyn, and emodin compounds, with strong anti-inflammatory properties [108]. Sohn et al. suggested a beneficial effect of the ethanol extract of the root in a DR murine model. It prevents diabetic-induced retinal vascular hyperpermeability, attenuating the HMGB1 signaling pathway through the downregulation of the RAGE-mediated activation of NF-κB. It directly blocks the binding of HMGB1 to RAGE, thus preventing retinal vascular inflammation. Moreover, fluorescein angiography demonstrated that PCE markedly inhibits fluorescein leakage, suggesting that it may prevent the breakdown of the BRB. PCE reduces the expression of HMGB1 in diabetic rat retinal tissue and inhibits the binding of NF-κB to the RAGE promoter, with considerable anti-inflammatory activity. It is worth mentioning that the oral administration of PCE showed no positive effects on glycemic and body weight control in the murine model [109].

### 4.4. Paeoniflorin

Paeoniflorin is a monoterpene glucoside extracted from the root of the *Paeonia Lactiflora*. Paeoniflorin shows anti-inflammatory properties and it is already used in traditional Chinese medicine for a wide range of pathologies [110,111]. Moreover, paeoniflorin shows immunomodulatory effects on microglial cells through the enhancement of the suppressor of cytokine signaling 3 (SOCS3) pathways [112]. Zhu et al. demonstrated, in an in vitro study on BV2 microglial cells exposed to high concentrations of glucose, that treatment with paeoniflorin reduces the expression of metalloproteinases-9 (MMP-9) through the inhibition of p38/NF-κB. In addition, paeoniflorin activates SOCS3, which blocks the TLR-4 pathway [112,113]. The repression of TLR-4 determines the reduction of HMGB1 mediated inflammation in retinal microglial cells [114].

### 4.5. Salicin

Salicin is the main component of the willow bark extract, commonly used in traditional medicine for its anti-inflammatory and antipyretic effects [115]. Salicin is metabolized to salicylic acid in vivo; therefore, it is also known as “nature’s aspirin”. Previous studies showed that salicin exerts protective effects on EC, both inhibiting angiogenesis [116] and reducing ROS production [117]. Song et al. demonstrated that the treatment of HRECs with salicin led to a reduction in HMGB1 release and the prevention of cellular apoptosis. Moreover, the authors demonstrated that salicin can suppress the production of IL-1β and its related cytokines, such as TNF-⍺ and IL-6, responsible for retinal toxicity. Salicin is also able to block the release of the adhesion molecules ICAM-1 and VCAM-1 and the NF-κB inflammatory pathway [118].

### 4.6. Ethyl Pyruvate (EP)

Ethyl pyruvate is a pyruvate derivative with the addition of an aliphatic ester group. EP is considered to be safer and more effective than pyruvate in inhibiting ROS and inflammation [119]. EP is a strong HMGB1 inhibitor. Treatment with EP promotes stable vascular growth and blocks retinal pathological neovascularization by preventing the overexpression of HMGB1. Moreover, EP can inhibit the expression of IL-6, TNF-⍺, and NF-kB, exerting a protective role in chronic inflammatory diseases such as DR [120].

### 4.7. Bradykinin (BK)

BK is a vasoactive peptide part of the kinins family. BK participates in several processes such as inflammation, pain, and cell proliferation [121]. Zhu et al. studied the effects of BK in HRECs exposed to high concentrations of glucose. Results show that BK can suppress oxidative stress and the release of inflammatory mediators. It can also control the process of neovascularization, downregulating the expression of VEGF. Lastly, BK inhibits the HMGB1/NF-κB signaling pathway, therefore controlling the growth and proliferation of HRECs [122].

### 4.8. Kallistatin

Kallistatin is an endogenous serine proteinase that plays numerous physiological and pathological roles like tumorigenesis, vasodilation, inhibition of neovascularization, inflammation, oxidative stress, cellular death, and fibrosis [123]. It stimulates the expression of eNOS, SIRT1, and SOCS3, while it inhibits VEGF, HMGB1, TNF-⍺, and NF-κB [124]. It has been demonstrated that vitreous humor levels of kallistatin in patients with PDR are lower when compared to healthy controls [125]. Xing et al. established that kallistatin is a strong inhibitor of angiogenesis and therefore may act as a potent drug in the prevention of PDR [126].

### 4.9. Compound 49b

Compound 49b is a recently discovered β-adrenergic receptor agonist that has already demonstrated efficacy in preventing apoptosis in in vitro models of EC and Müller cells exposed to high glucose [127,128]. Recent evidence shows that compound 49b can inhibit HMGB1 expression, TLR-4 downstream signaling, and, therefore, NF-κB in both EC and Müller cells. This leads to the idea that this agonist may preserve vascular and neuronal integrity in the diabetic retina [126,129].

### 4.10. Cyclosporine A (CyA)

Cyclosporine A is a polypeptide derived from the fungi *Beauveria nevus* and *Tolypocladium inflatum*, and it is well known for its anti-inflammatory and immunosuppressive effects [130]. Wang et al. demonstrated that CyA attenuates the enhanced expression of IL-1β and TNF-⍺ in the retinas of diabetic rats, probably via the suppression of HMGB1. The intravitreal injection of CyA may represent a novel therapeutic strategy to treat DR [131].

CyA is also involved in the reduction of BRB permeability in diabetic rats. In particular, it reduces the levels of IL-1β, nitric oxide (through a decreased expression of iNOS), and IL-1β-induced cyclooxygenase-2 (COX-2) expression. Moreover, CyA decreases vitreous protein concentration in diabetic rats. The authors suggest that this reduction in vitreous protein concentration can be linked to the reduction of BRB permeability [132].

## 5. Conclusions

The purpose of this review is to show the role of HMGB1 in DR. Many studies have demonstrated that DM, and then hyperglycemia, upregulate the expression of and increase in the levels of HMGB1. This situation activates several pathways and involves a large number of molecules, such as ERK, NF-κB, ICAM, RAGE, VEGF, and TLR. The final result is the activation and increase of inflammation, angiogenesis, oxidative stress, and, ultimately, retinal damage to the patients. At the same time, the involvement of this great number of molecules can provide a hint of new therapeutic approaches to be developed and studied.

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
