# Peer review of "The Complex Relationship between Diabetic Retinopathy and High-Mobility Group Box: A Review of Molecular Pathways and Therapeutic Strategies"

_antioxidants, 2020, doi:10.3390/antiox9080666_

Round 1
Reviewer 1 Report
The manuscript by Nebbioso et al. provides a thorough account of the current knowledge regarding the involvement of HMGB1 in diabetic retinopathy, and suggests some potential future treatment options that are currently under investigation. I believe this article will be valuable to the scientific research community in the field of diabetic retinopathy. The following minor revisions will improve the quality of the manuscript:
1) Revision for English grammar and style is strongly recommended.
2) In fig. 1, the “altered external retina” arrow should be pointed the opposite way to be consistent with the rest of the figure.
3) In Scheme 1, under “hyperglycemia”, it should be “apoptosis” instead of “apoptotic”.
4) The “HMGB1 and DR” section is long and contains lots of information. It would really be helpful to include a diagram to summarize and illustrate the multiple functions of HMGB1 in DR.
5) The “future therapeutic approaches” section is interesting, though it may be important to discuss the potential drawbacks of the suggested therapeutic compounds if known.
Author Response
1) Revision for English grammar and style is strongly recommended.
We have corrected the errors.
2) In fig. 1, the “altered external retina” arrow should be pointed the opposite way to be consistent with the rest of the figure.
We have changed the arrows.
3) In Scheme 1, under “hyperglycemia”, it should be “apoptosis” instead of “apoptotic”.
We have corrected the word.
4) The “HMGB1 and DR” section is long and contains lots of information. It would really be helpful to include a diagram to summarize and illustrate the multiple functions of HMGB1 in DR.
Ok, we've put in a diagram
5) The “future therapeutic approaches” section is interesting, though it may be important to discuss the potential drawbacks of the suggested therapeutic compounds if known.
Unfortunately, there are no works on human samples useful for the purpose of the Reviewer's request and the discussion on the topic is very limited.
Reviewer 2 Report
The review manuscript titled The complex relationship between diabetic retinopathy and High-Mobility 2 Group Box: a review of molecular pathways and therapeutic strategies is a well-structured presentation of the findings published by various different research programs. The review is well-written. However, including complementary tables and diagrams/illustrations would significantly enhance reader convenience. To that end, several suggestions are provided below.
Figures 1 1nd 2 do not add greatly to the understanding of the molecular pathways or therapeutic strategies that are the main designs of this review, as indicated by the title. The figures should be removed.
A drawing of the structure of HMGB-1 indicating the various domains as they pertain to roles in DR would be helpful.
The authors should provide summary illustrations showing relationships of molecules activated in the various different cell signaling pathways involving HMGB-1 in DR. Explanations of the possible cross talk of different pathways, which promote HMGB-1-mediated DR advancement, should be provided in the illustration legends or text of the review.
A table listing the HMGB-1 inhibitors that have been tested in human and animal trials, and on cultured cells, would be a meaningful addition. The table might indicate the drug/chemical tested, patient/animal gender and age (or type of cells in culture), approach to diabetic inducement (e.g., STZ), the mechanism of action and results of the treatment, and references to the studies.
Abbreviations used for non-proliferative DR and proliferative DR.
Author Response
1) Figures 1 1nd 2 do not add greatly to the understanding of the molecular pathways or therapeutic strategies that are the main designs of this review, as indicated by the title. The figures should be removed.
We have inserted 2 figures to show the serious damage caused by diabetic disease and the first Reviewer asked us to modify one figure. We leave to the Reviewer 2 and Editor the choice of removing or no the 2 figures.
2) A drawing of the structure of HMGB-1 indicating the various domains as they pertain to roles in DR would be helpful.
We have inserted the figure as recommended by the Reviewer.
3) The authors should provide summary illustrations showing relationships of molecules activated in the various different cell signaling pathways involving HMGB-1 in DR. Explanations of the possible cross talk of different pathways, which promote HMGB-1-mediated DR advancement, should be provided in the illustration legends or text of the review.
We added Schedule 2 as requested by the Reviewers.
4) A table listing the HMGB-1 inhibitors that have been tested in human and animal trials, and on cultured cells, would be a meaningful addition. The table might indicate the drug/chemical tested, patient/animal gender and age (or type of cells in culture), approach to diabetic inducement (e.g., STZ), the mechanism of action and results of the treatment, and references to the studies.
We added a Table 1 as requested by the Reviewer.
5) Abbreviations used for non-proliferative DR and proliferative DR.
We have corrected